# Glaucoma Incidence and Progression in Diabetics: The Canary Islands Study Using the Laguna ONhE Application

**DOI:** 10.3390/jcm11247294

**Published:** 2022-12-08

**Authors:** Marta Gonzalez-Hernandez, Daniel Gonzalez-Hernandez, Nisamar Betancor-Caro, Isabel Guedes-Guedes, Morten Kirk Guldager, Manuel Gonzalez de la Rosa

**Affiliations:** 1INSOFT S.L. 25 de Julio, 34, 38004 Santa Cruz de Tenerife, Spain; 2Ophthalmology Department, Hospital Universitario de Canarias, Carretera Ofra S/N, 38320 San Cristobal de La Laguna, Spain; 3Ophthalmology Department, Hospital Insular de Gran Canaria, Avenida de Canarias, S/N, 35016 Las Palmas de Gran Canaria, Spain; 4RetinaLyze System A/S, Sankt Lukas Stiftelsen, Bernstorffsvej 20, 2900 Hellerup, Denmark

**Keywords:** optic nerve head, glaucoma, progression, diabetes, fundus photographs, optic disc

## Abstract

Background: Laguna ONhE provides a globin distribution function (GDF), in which a glaucoma discriminator based on deep learning plays an important role, and there is also an optimized globin individual pointer (GIP) for progression analysis. Methods: Signs of optic nerve glaucoma were identified in 1,124,885 fundus images from 203,115 diabetics obtained over 15 years and 117,813 control images. Results: A total of 743,696 images from 313,040 eyes of 173,661 diabetics were analysed. Some exclusions occurred due to excessive illumination, poor quality, or the absence of optic discs. Suspicion of glaucoma was reported in 6.70%, for an intended specificity of 99% (GDF < −15). More signs of glaucoma occur in diabetics as their years of disease increase, and after age 60, compared to controls. The GIP detected progression (*p* < 0.01) in 2.59% of cases with 4 controls and in 42.6% with 14 controls was higher in cases with lower GDF values. The GDF was corrected for the disc area and proved to be independent of it (r = 0.001925; *p* = 0.2814). Conclusions: The GDF index suggests a higher and increasing glaucoma probability in diabetics over time. Doubling the number of check-ups from four to eight increases the ability to detect GIP index progression by a factor of 5.

## 1. Introduction

Whether diabetes mellitus (DM) is a risk factor for the development of primary open-angle glaucoma (POAG) remains controversial, although it has been suggested in the Blue Mountains study [1] and other studies [2]. A meta-analysis of 13 papers found considerable heterogeneity among the case-control studies and no significant heterogeneity among the cohort studies, leading to the conclusion that DM is indeed a risk factor for glaucoma [3].

To facilitate the diagnosis of diabetic retinopathy in a population dispersed among islands, since 2006, the Canary Islands Health Service has maintained a network called Retisalud, made up of more than 60 centres. Each of them has fundus cameras networked with the main hospitals. Trained technicians obtain fundus images of diabetic patients approximately every year. Images suggestive of retinopathy are referred to the hospitals and evaluated by ophthalmology specialists. Retisalud’s general doctors keep patients without significant retinopathy in their database. Those requiring strict control (moderate non-proliferative retinopathies) or specific treatment (severe non-proliferative retinopathies, proliferative retinopathies, and macular oedemas) are monitored by ophthalmologists in their areas and hospitals. The availability of this wealth of information encouraged us to try to use it to confirm the relationship between the two diseases.

Laguna ONhE is a method for the colourimetric evaluation of the distribution of haemoglobin at the optic nerve, which has been used since 2013 [4], mainly for the study of different forms of chronic glaucoma [5,6,7] but also congenital glaucoma [8]. The application has been compared with other diagnostic methods, such as confocal scanning laser ophthalmoscopy (HRT) [9], scanning laser polarimetry (GDx), optical coherence tomography (OCT) [10], angio-OCT [11], and perimetry [12]. In particular, its reproducibility [13], its usefulness for the diagnosis of early glaucoma [14], and its empowerment upon associating its information with the irregularity of the visual field have been demonstrated [15].

In recent years, the application has been fully automated using a variety of procedures. In particular, it has been provided with several artificial intelligence mechanisms, including deep learning convolutional neural networks, to segment the edges of the optic disc, identify its laterality, determine the quality of the image, identify the presence or absence of the entire optic disc and its surroundings, achieve the segmentation of its vessels, and identify the existence of normal or glaucoma characteristics, both in the optic disc and in its surroundings [15].

Other algorithms are responsible for recognizing whether the image has been zoomed in and estimating the disc area [16], the area of the cup, and the area of the rim sectors. The nerve tissue is also compared by colourimetry with the vessels of the central retinal artery and vein branches, which are used as a standard. The relationship between them allows the estimation of the presence of haemoglobin in each area of the optic disc. In particular, the distribution of haemoglobin in the inferior and superior zones of the nerve in relation to the nasal and temporal zones, and the relationships between the size and shape of the cupping, weighted according to the size of the optic disc, as well as other factors such as the presence of peri-papillary atrophies, are synthesized into two main indices: the globin or glaucoma discriminant function (GDF), in which the deep learning glaucoma vs. normal classifier has a high influence, with an approximate range of −100 to +85 units, and the globin individual pointer (GIP), which is more focused on the particular characteristics of the subject’s own nerve, which allows temporal changes to be detected due to its greater reproducibility [17], with an approximate range of −100 to +100.

Additionally, Laguna ONhE has been applied for glaucoma screening in optometry and ophthalmology centres, mainly in Central Europe and Scandinavia, by the company RetinaLyze System A/S (Hellerup, Denmark), who have analysed more than 840,000 images of optical discs to date.

The large number of images accumulated by Retisalud over 15 years led us to promote a research project in three consecutive phases. In the first phase [18], the reduction in vessels in diabetic optic discs with and without signs of glaucoma was tested. In the second phase, which is described in the current paper, a retrospective study of all the accumulated images was performed. In the third phase, volunteer patients were recruited and underwent complimentary glaucoma-oriented examinations for diagnostic verification.

The main aims of the study can be summarized in four aspects. The first is to assess the suggested relationship between diabetes and glaucoma in a large population sample. The second is to confirm the usefulness of the Laguna ONhE method as an automatic mass screening procedure for glaucoma. The third is to evaluate the ability of Laguna ONhE to detect longitudinal changes in nerve perfusion and vascularization related to age, glaucoma, or diabetes. The fourth is to examine whether it is possible to achieve glaucoma screening indices that are relatively independent of the optic disc size.

## 2. Materials and Methods

### 2.1. ONH Hb Measurements

In summary, the Laguna ONhE method locates the position of the optic nerve in each image, identifies the inner edge of the Elschnig scleral ring, and segments the optic disc vessels. The presence of haemoglobin is then topographically estimated in the optic nerve tissue using the colour of the vessels as a reference. Haemoglobin mainly absorbs green radiation and reflects most of the red. Therefore, the reference colour of the vessels was calculated using the values of the red (R) and green (G) channels of their pixels by the formula (R − G)/R. The same equation was used for the pixels of the tissue, and finally, the result was expressed as a percentage. Estimates of the cup size and position were also obtained, and the results of the cup, rim sectors, vertical cup/disc ratio, and cup/disc area ratio were compared with the percentiles achieved in the normal population.

For a detailed description of the method and the artificial intelligence tools used, please refer to the appendix in a previous publication [15].

### 2.2. Laguna ONhE Indices

The GDF index combines, in a single value, data relevant to distinguish normality from glaucoma. These are the vertical C/D ratio, estimated from the haemoglobin distribution, and the percentiles of the estimated haemoglobin values relative to the normal population values, obtained in 24 areas of the nerve, weighted by the disc area. All these data, except the disc area which is constant, are continuous variables. A deep learning classifier also plays an important role. This is a practically dichotomous variable, accumulating values close to 1 (normal) and 0 (glaucoma). In obvious cases for the neural network, the classifier remains stable near 1 or 0, but in borderline cases, it can change from 0 to 1 or vice versa between two examinations. For this reason, GDF is a useful index for diagnosis, but not for follow-up. GIP is a similar index, but with minimal weight given to the deep learning classifier. It is therefore less diagnostic, but more stable, making it easier to detect changes for progression studies.

### 2.3. Subjects

The study protocol of this cross-sectional retrospective study adhered to the principles of the 1975 Declaration of Helsinki revised in 2013 and was approved by the Research Ethics Committee of the Hospital Universitario de Canarias (CHUC_2018_09 (V4)). Retisalud provided the images in a completely anonymous format. Given this fact, the purpose of the work, and the volume of images, the committee waived the requirement for informed consent from the patients. Consent was obtained from all the participants for the images provided by RetinaLyze System A/S for the control group.

As explained in the introduction, the Retisalud programme maintains in its database only patients without retinopathy or with signs of very initial retinopathy (isolated microaneurysms or exudates). Cases requiring specific treatment (macular oedema, neovascularisation, ischaemic conditions, etc.) were transferred to specialized centres and were not included in this series.

Retisalud provided 1,124,885 fundus images obtained from 203,115 diabetic patients of the Canary Islands between 18 May 2006 and 31 March 2021. RetinaLyze System A/S provided a series of 117,813 control images, randomly selected from the general population, to obtain a balanced age distribution with respect to the diabetic series. To avoid selection bias, these images were obtained in subjects attending for a general assessment, not specifically for glaucoma.

### 2.4. Statistical Analyses

The clinical statistical analyses were performed using the Excel 2016 program (Excel Microsoft Corp., Redond, WA, USA) and MedCalc (Version 20.110—64 bits; MedCalc Software bvba, Mariakerke, Belgium).

## 3. Results

Retisalud diabetic series: A total of 743,696 images from 313,040 eyes (160,443 or 51.3% of left eyes and 152,597 or 46.7% of right eyes) from 173,661 patients (82,538 or 47.5% female and 91,123 or 52.5% male, with a mean age of 64.3 years at the last examination; sd = 12.9) were able to be analysed (66.1% of the total images). Both eyes could be analysed in 139,379 cases. The main reasons for not being able to analyse the images were excessive illumination (37.6%), low image quality (28.4%), the optic nerve being absent (10.7%), the optic nerve being sectioned (3.9%), or the optic nerve being very close to the edge (19.4%). The average number of examinations per eye was 2.38 (sd = 1.85), ranging from 1 to 23.

The mean duration of the follow-up of all the diabetics was 2.67 years (sd = 3.26), with a range between 0 and 14.53 years.

Of the eyes examined on their last date, 6.70% had a GDF below −15 (intended specificity: 99%) (Figure 1).

Both eyes were analysed in 139,379 patients (80.3%), and one eye, in 34,282 (19.7%). A significant relationship between the GDF values of both eyes was observed (r = 0.648; *p* < 0.0001; regression analysis). In the left eye, 6.44% of the cases had a GDF less than −15, and in the right eye, 6.72% did. Both eyes had GDF <−15 in 2.83% of cases, and 2.72% had one eye with GDF <−15 and the other with a GDF between −15 and 0 (intended specificity: 95%). When, in one eye, GDF values lower than −15 were observed in the contralateral eye; generally low values were also found (mean = −11.85; sd = 23.86).

The control series provided by RetinaLyze System A/S had a mean age of 64.42 years (sd = 16.83; *p* = 0.49; Student’s *t*-test). The frequency of cases with GDF values below −15 was similar in the control group and in the diabetic group before the age of 60 years. From this age onwards, the frequency of these values increased significantly in diabetics, including those over 90 years of age, although, in this group, the differences were not significant because of the limited number of cases (Figure 2).

For 73,106 eyes, images obtained between 4.5 and 5.5 years after the first examination were available, i.e., those with approximately 5 more years of the evolution of diabetes from the initial examination. At the first examination, the patients had a mean age of 59.07 years (sd = 11.19), and at the last examination, 63.88 years (sd = 11.19). The percentage of cases with GDF <−15 was initially 3.93%; 4.81 years later, this percentage was 5.47%.

The frequency distribution of the disc area was slightly asymmetric. The mean was 1.93 mm^2^ (sd = 0.35) and the median was 1.90 mm^2^. The influence of the disc area on the GDF index, analysed in the first control, was not significant (r = 0.0019; *p* = 0.28) (Figure 3A).

However, a significant relationship between the GDF index and vertical cup/disc ratio was observed (r = −0.73; *p* < 0.0001) and also between the disc area and vertical cup/disc ratio (r = −0.30; *p* < 0.0001).

Figure 4 shows an example of the assessment of the progression of the GIP index and the areas of the cup and rim sectors.

In cases where at least 4 examinations were performed, progression could be observed in 2.59%, and when there were 10 or more examinations, it was detected in 19.54–42.6%, with a probability of *p* < 0.01 for the corresponding regression line. As the number of check-ups increased, smaller changes of about 3 GDF units per year and increases in the percentage cup size of less than 1% could be detected (Table 1).

Figure 5 shows an example of changes in the haemoglobin distribution in the optic nerve of a patient over 11.04 years.

## 4. Discussion

The prevalence of diabetes in European countries is estimated at 4–7% [19] but is higher in Spain [20] and especially in the Canary Islands, presumably for genetic and dietary reasons. It should be borne in mind that patients under control by Retisalud represent more than 9% of the total population of the islands, of any age. Severity referrals and undiagnosed cases would undoubtedly increase this estimate substantially, especially in adults.

The importance of the diagnosis and control of retinopathy is unquestionable. This paper suggests that a structure such as Retisalud, programmed for this purpose, can be used for the simultaneous control of glaucoma, or even for the screening of the general nondiabetic population.

The false-positive rate of Laguna ONhE is likely to be at least 1%, as the specificity has been estimated to be close to 99% for GDF < −15 [15]. Consequently, the true incidence of possible glaucoma shown in Figure 2 and the remaining results should be corrected downwards, so that it would be exceptional below the age of 30 years. Still, the observed frequency could be at the upper end of the expected range, which is assumed to be between 2.1 and 5.8% [21]. This higher frequency may be largely due to the relatively advanced age of the selected subjects, as the incidence of glaucoma increases with age, and, on the other hand, due to the coincidence with diabetes.

We have previously published that there is a reduction in the vascular component of the optic disc in long-term controlled diabetic patients, as determined by counting the pixels identified as vascular by our method [18]. These data, together with other information obtained by angio-OCT on reductions in microvascularization in various regions of the ocular tissue in glaucoma, whether in the peripapillary vessels [22], inside the optic disc [23], in the whole-image vessel density [24], or in the macular region [25,26], also confirm that diabetes contributes, in some way, to glaucomatous damage.

A limitation of this study is that a reference sample in a population identical in size and genetic characteristics to the diabetic population was not available. However, both samples were composed of Caucasian individuals. The population of the Canary Islands has its main origin in individuals from the Iberian Peninsula, although there is also a considerable proportion from other European origins. The absence of significant differences in the signs of glaucoma in subjects under 60 years of age suggests that, most probably, the differences observed at older ages are due to diabetes and not to racial particularities.

The results we have obtained in the reference population suggest that diabetes microangiopathy participates in or facilitates the appearance of defects comparable to glaucomatous defects in patients with a long history of diabetic disease. The difference obtained in this study may be an underestimate, as the general reference population should also include diabetics in the usual minority proportion.

GIP progression data are obviously more important for subjects with GDF values suggestive of glaucoma, but to a lesser extent, we have also observed them to be frequent in diabetic subjects without suspected glaucoma. GIP is mainly sensitive to haemoglobin reduction in the inferior and superior sectors of the optic disc, and, in previous studies, we did not observe that age significantly altered the haemoglobin density [5]. It, therefore, seems likely that the worsening of this index with age is related to diabetic microangiopathy and its relationship with glaucoma.

Finally, our results indicate that the GDF index provided by the Laguna ONhE is very weakly dependent on the optic disc area and is not confounded by glaucoma in macropapillae, as can occur with other morphological indices, such as the vertical cup/disc ratio and rim area.

Despite this study being based on previously widely validated criteria, the main limitation of this study is that there was no diagnostic confirmation of each case. A prospective study of a sample of the patients from this study is currently underway to verify the degree of validity of these diagnostic suspicions.

## 5. Conclusions

This study aims to corroborate the idea that diabetes is a considerable risk factor for glaucoma, suggesting that the incidence in these patients is higher than in the general population, and demanding special attention, especially in older patients.

The translational relevance of this study lies in the fact that it has managed to demonstrate that health structures designed for the control of diabetes can be used for the diagnosis and control of glaucoma. Particularly important is the fact that these health facilities can, in this manner, extend their scopes of action to nondiabetic populations, with a consequent increase in efficiency.

In the current study, we have found that at least two-thirds of the images obtained for the follow-up of diabetic retinopathy could be used for the evaluation of glaucoma, using an automated method of optic disc analysis. However, the practical effectiveness could be much higher, since the adequate training of technicians would greatly increase the number of useful images. This could be achieved, on the one hand, by avoiding overexposure of the images, but also by controlling for the complete presence of the optic disc and using neural networks that check the quality of the images, in a program such as the one we have used, so that the user repeats the image capture if necessary until a truly useful image is obtained.

## Figures and Tables

**Figure 1 jcm-11-07294-f001:**
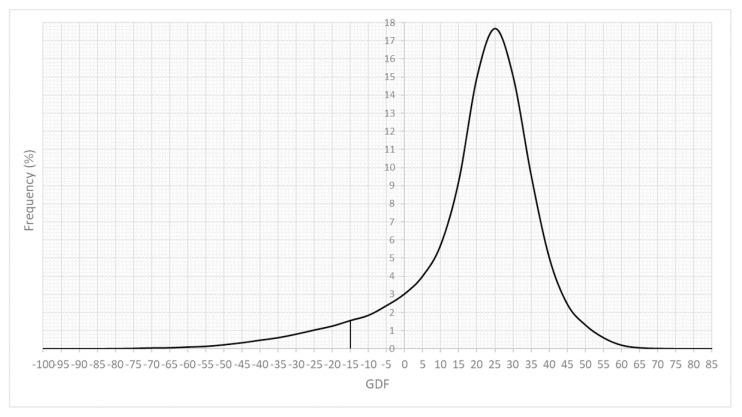
Frequency distribution of the GDF index at the last examination of each eye. Zero is approximately the minimum value in 5% of the normal population, and −15 is the value in 1% of the normal population.

**Figure 2 jcm-11-07294-f002:**
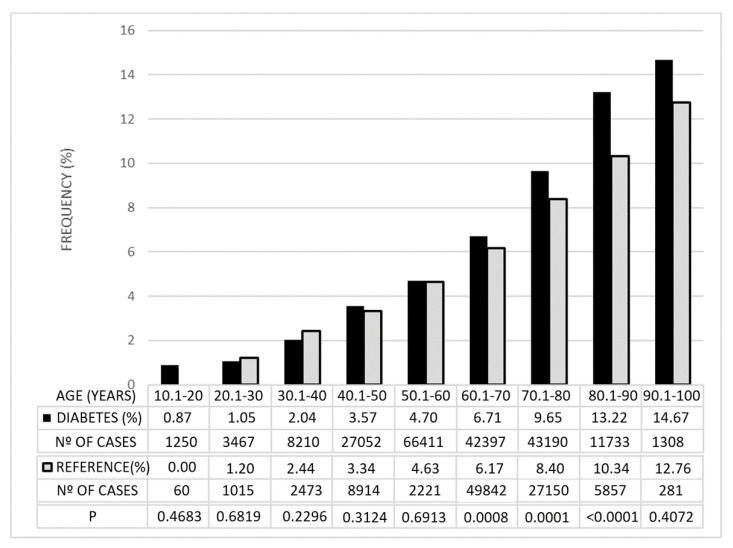
Percentages of cases with GDF <−15 in the diabetic group and in the reference group. They are grouped by age and the degree of statistical significance (*p*) of the differences observed between the two groups.

**Figure 3 jcm-11-07294-f003:**
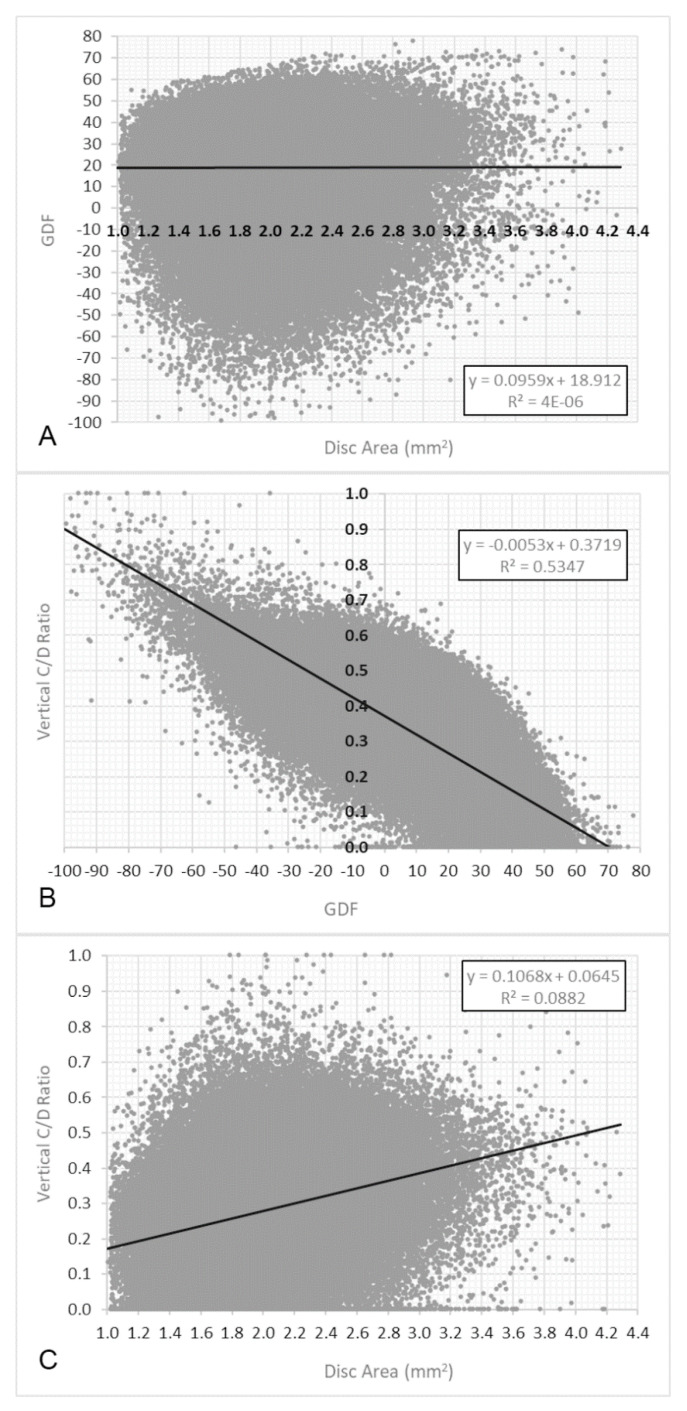
(**A**). Relationship between GDF and disc area values in the first control image. (**B**). Relationship between GDF and vertical C/D ratio. (**C**). Relationship between disc area and vertical C/D ratio.

**Figure 4 jcm-11-07294-f004:**
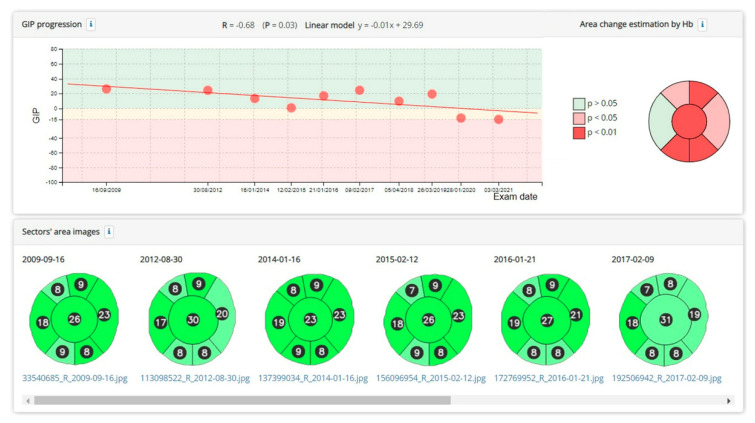
Example of a significant reduction in the GIP index over several years. A significant increase in the percentage area of the cup and a reduction in several sectors of the rim is also observed. The change from dark green to light green in the optical disc sectors indicates that the haemoglobin density, compared to the normal population, has decreased from a value above the normal median to a value below the median.

**Figure 5 jcm-11-07294-f005:**
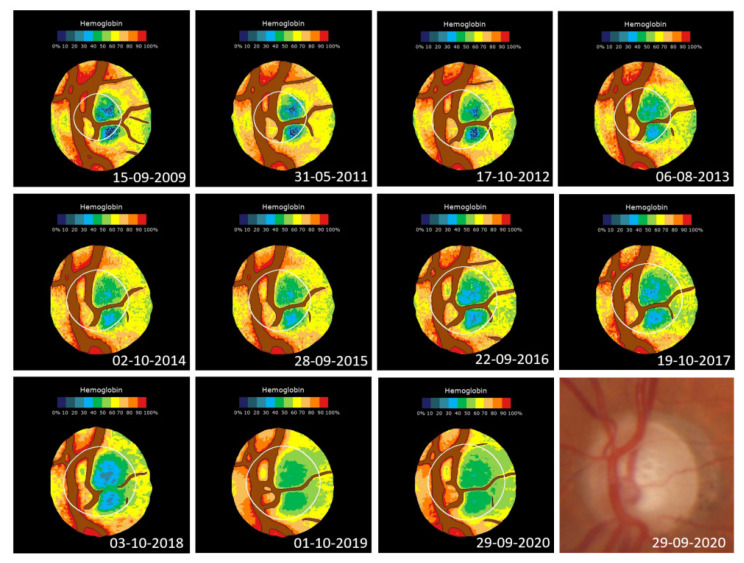
Changes in the haemoglobin distribution in a 1.74 mm^2^ optic nerve of a diabetic patient with associated glaucoma. From 2009 to 2020, the GIP index decreased from an initial value of 43.2 to −42.5 (r = 0.94; *p* < 0.0001) and the cup went from occupying 20.0% of the disc to 53.1% (*p* = 0.97; *p* < 0.0001). The GDF index goes from an initial borderline situation (−0.49) to clearly anomalous (−52.0). Bottom right: last original image.

**Table 1 jcm-11-07294-t001:** Cases with significant progression (*p* < 0.01). The percentage of cases that progressed, the slope per year of the GIP index, and the percentage area of the cup in relation to the number of control examinations performed are shown.

Number of Exams	4	5	6	7	8	9	10	11	12	13	14
Number of eyes	25,020	15,221	9411	5495	3349	1950	1090	565	294	131	61
Cases with progression (%)	2.59	4.96	7.26	10.94	13.38	15.23	19.54	25.66	21.8	33.6	42.6
GIP slope/year (mean)	−3.59	−3.8	−3.65	−3.23	−3.01	−2.98	−2.81	−2.66	−2.3	−2.8	−2.4
GIP slope/year (sd)	3.93	3.32	2.95	2.68	2.35	2.06	1.66	1.55	1.8	1.4	1.2
Cup area (%) slope/year (mean)	−1.03	−1.1	−1.03	−0.92	−0.74	−0.86	−0.77	−0.82	−0.5	−0.7	−0.6
Cup area (%) slope/year (sd)	1.45	1.16	1.09	1.06	0.88	0.75	0.71	0.62	0.7	0.7	0.5

## Data Availability

The data presented in this study are available on request from the corresponding author. The data are not publicly available due to ethical restrictions.

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
