# Peer review of "Glaucoma Incidence and Progression in Diabetics: The Canary Islands Study Using the Laguna ONhE Application"

_jcm, 2022, doi:10.3390/jcm11247294_

Round 1

Reviewer 1 Report

The article was interesting to read but a bit hard to follow, and that might be because Laguna OnhE is not a ubiquitous device. Therefore, the definitions of GDF and GIP should be thoroughly explained. 

Another issue is in reference to the first aim of this paper: assessing the suggested relationship between diabetes and glaucoma in a large population sample. There is information about the average duration of diabetic patients, but no information to the extent of seriousness of DM was not mentioned.  Considering the fact that patients with poorly controlled DM are likely to develop secondary glaucoma with uncontrolled increased intraocular pressure, the status of diabetic patients should be stated. 

Mentioned above already, Laguna ONhE is not one of conventional devices for glaucoma surveillance and diagnosis, I am a bit skeptical about its usefulness as a glaucoma diagnostic tool. Please provide more references to convince readers like me. 

Author Response

1.- The article was interesting to read but a bit hard to follow, and that might be because Laguna OnhE is not a ubiquitous device. Therefore, the definitions of GDF and GIP should be thoroughly explained.

Answer: We thank the reviewer for his comments. It is indeed more difficult to assimilate novelties than to read "fractal-papers". The following sentence has been added as section 2.2 to Materials and Methods: “The GDF index combines, in a single value, data relevant to distinguish normality from glaucoma. These are the vertical C/D ratio, estimated from the haemoglobin distribution, and the percentiles of the estimated haemoglobin values relative to the normal population values, obtained in 24 areas of the nerve, weighted by the disc area. All these data, except the disc area which is constant, are continuous variables. A Deep Learning classifier also plays an important role. This is a practically dichotomous variable, accumulating its values close to 1 (normal) and 0 (glaucoma). In obvious cases for the neural network the classifier remains stable near 1 or 0, but in borderline cases, it can change from 0 to 1 or vice versa between two examinations. For this reason, GDF is a useful index for diagnosis, but not for follow-up. GIP is a similar index, but with minimal weight given to the Deep Learning classifier. It is therefore less diagnostic, but more stable, making it easier to detect changes for progression studies.”

2.- Another issue is in reference to the first aim of this paper: assessing the suggested relationship between diabetes and glaucoma in a large population sample. There is information about the average duration of diabetic patients, but no information to the extent of seriousness of DM was not mentioned.  Considering the fact that patients with poorly controlled DM are likely to develop secondary glaucoma with uncontrolled increased intraocular pressure, the status of diabetic patients should be stated.

Answer: The degree of severity of MD has been reported twice in the text. Specifically:

Pag 1-2. Lines 44-45. “Retisalud's general doctors keep patients without significant retinopathy in their database. Those requiring strict control (moderate non-proliferative retinopathies) or specific treatment (severe non-proliferative retinopathies, proliferative retinopathies, and macular oedemas) are monitored by ophthalmologists in their area and hospitals”.

Pag 3. Lines 121-125. “As explained in the Introduction, the Retisalud programme maintains in its database only patients without retinopathy or with signs of very initial retinopathy (isolated microaneurysms or exudates). Cases requiring specific treatment (macular oedema, neovascularisation, ischaemic conditions, etc.) were transferred to specialised centres and were not included in this series.”

We emphasize: Patients in advanced stages of the disease, for example neovascularisation, which could be associated with secondary glaucoma, have not been included. All these patients are referred to specialised centres and excluded from the series monitored by Retisalud.

3.- Mentioned above already, Laguna ONhE is not one of conventional devices for glaucoma surveillance and diagnosis, I am a bit skeptical about its usefulness as a glaucoma diagnostic tool.

Answer: Having a sceptical attitude is excellent for facing the unknown, keeping an open mind to new ideas. However, it cannot be an argument for stating an opinion and taking a position. Science would not progress if only the conventional and known were accepted. Medicine in particular has a strong tendency to inertia, out of a logical need for caution. To convince a sceptic, the best option is to provide him with personal experience. We have created a reviewer login on our test server, which you will find at the following address “https://test-laguna.insoft.es/login/”. Using your user ID and password “[email protected]”, you can analyse your own fundus images and come to a judgement based on experience. It is quite easy to use. A user manual is enclosed. If you would like to keep this access in the future for research purposes, we will be happy to keep the access active in the future.

4.- . Please provide more references to convince readers like me.

Answer: The absence of previous references cannot be a criterion for evaluating a paper, even if it helps. If it were, there would be the paradox that no paper with a new idea could be published. But this is not the case with this paper. There have been publications on the subject since 2013. 14 of them have been cited in this paper and have been published in prestigious journals. This does not seem a negligible number. We understand that not all of them may be easily accessible to the reviewer. We have therefore put them together in a zipped file that has been associated with this response.

Reviewer 2 Report

This is very informative manuscript.  However, there are some issues that need to be addressed and improved.

1. The reference #17 used for GIP index is an abstract, therefore not providing much about GIP.  More explanation or a proper reference for GIP is needed.

2. The title seems to be incorrect for the reference #18.

3. It’s hard to see the relevance of Figure 3 with the main finding in line 178-179 ‘However, a significant relationship between the disc area and vertical cup/disc ratio 178 was observed (r=-0.73; p<0.0001).’.   I would recommend adding another Figure to back up the statement in line 178-179, making Figure 3(A) and 3(B).

4. There are two ‘Figure 5’ entries.  The heading and explanation for ‘Figure 5’ seems to be separated, therefore need to be tidy up. 

Author Response

This is very informative manuscript.  However, there are some issues that need to be addressed and improved.

1.- The reference #17 used for GIP index is an abstract, therefore not providing much about GIP.  More explanation or a proper reference for GIP is needed.

Answer: We thank the reviewer for his comment. A more detailed description of the GDF and GIP indices has been included in the response to reviewer 1's first question and added as section 2.2 of Materials and Methods.

2.- The title seems to be incorrect for the reference #18.

Answer: Thank you very much for the observation. We have made the corresponding corrections.

  1. It’s hard to see the relevance of Figure 3 with the main finding in line 178-179 ‘However, a significant relationship between the disc area and vertical cup/disc ratio 178 was observed (r=-0.73; p<0.0001).’. I would recommend adding another Figure to back up the statement in line 178-179, making Figure 3(A) and 3(B).

Answer: Thank you very much for the observation. The reviewer's suggestion has enabled us to detect and correct an error in figure 3. We have replaced it with a triple figure (A, B, C). For consistency with the text, the position of the figure has been moved back and its legend has been enlarged.

  1. There are two ‘Figure 5’ entries. The heading and explanation for ‘Figure 5’ seems to be separated, therefore need to be tidy up.

Answer: There is no double entry. The sentence "Figure 5 shows an example of changes in the haemoglobin distribution in the optic nerve of a patient over 11.04 years" is part of the main text and was located above the figure. It is now at the end of page 7. Figure 5’s legend is located, as usual, with smaller character size, bellow the figure.

Round 2

Reviewer 1 Report

The authors  elaborated on my questions.

Thank you. 

Reviewer 2 Report

Authors addressed all issues I pointed.